# Agent-Based and System Dynamics Modeling of Water Field Services

## Bernard Amadei 

Department of Civil Engineering, University of Colorado, Boulder, CO 80309-0428, USA; amadei@colorado.edu; Tel.: +1-303-929-8167

**Abstract:** This paper explores the applicability of the agent-based (AB) and system dynamics (SD) methods to model a case study of the management of water field services. Water borehole sites are distributed over an area and serve the water needs of a population. The equipment at all borehole sites is managed by a single water utility that has adopted specific repair, replacement, and maintenance rules and policies. The water utility employs several service crews initially stationed at a single central location. The crews respond to specific operation and maintenance requests. Two software modeling tools (AnyLogic and STELLA) are used to explore the benefits and limitations of the AB and SD methods to simulate the dynamic being considered. The strength of the AB method resides in its ability to capture in a disaggregated way the mobility of the individual service crews and the performance of the equipment (working, repaired, replaced, or maintained) at each borehole site. The SD method cannot capture the service crew dynamics explicitly and can only model the average state of the equipment at the borehole sites. Their differences aside, both methods offer policymakers the opportunity to make strategic, tactical, and logistical decisions supported by integrated computational models.

**Keywords:** agent-based method; system dynamics; water field services

---

## 1. Introduction

Many problems residing at the crossroads between socio-economic, natural, and infrastructure systems are complex and cannot be addressed with simple analytical tools. Computer simulations can be used instead to decide where to intervene in these systems while accounting for their evolution over time (i.e., their dynamics). This cannot be done blindly, and a methodology must be followed, starting with getting acquainted with the context and scale of the landscape in which the problems unfold. This is followed by developing a clear understanding of the issues and their dynamics, being able to simplify these issues, and accessing databases. Once simplified and abstracted, the next stage is to select appropriate modeling tools jointly with realistic input data to simulate and reproduce the dynamics of the issues of interest. Finally, the last step is to carry out various sensitivity analyses, recommend possible intervention scenarios, and decide on the pros and cons of their implementation.

Regardless of how one approaches the modeling of complex problems, it must be remembered that models are based on an interpretation of reality and are not reality itself. They are virtual representations needed to simplify the complexity of the world around us. As remarked by George Box, "essentially all models are wrong, but some are useful [ . . . ] the approximate nature of the model must always be borne in mind" [1]. In their interpretation of reality, models must be useful, comprehensive, and sound enough to be able to "reason, explain, design, communicate, act, predict, and explore" [2].

Since the 1940s, a variety of modeling tools have been proposed in various disciplines of systems and complexity science to address complex problems (see the map by Castellani [3]). Among these tools, Borshchev and Filippov [4] consider three types: system dynamics (SD), discrete event modeling

(DE), and agent-based (AB) modeling. As shown in Figure 1, these three categories address different levels of abstraction and details and can be used at different levels of decision making (i.e., strategic, tactical, and operational).

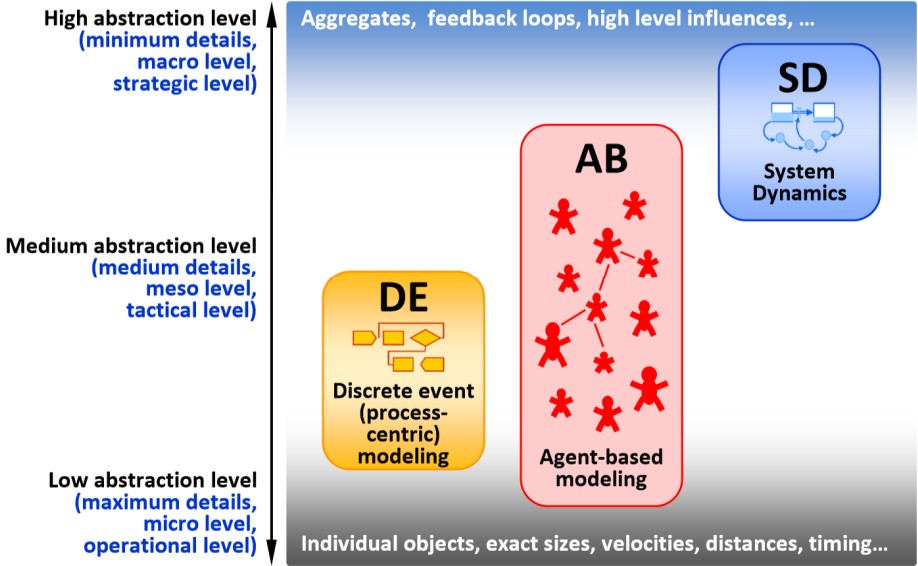

**Figure 1.** Categories of system modeling tools and levels of abstraction (from Borshchev and Grigoryev [5]).

In a nutshell, the SD method is top-down and can be applied to systems with high levels of aggregation (i.e., high abstraction). The method assumes that all processes are continuous and considers multiple feedback mechanisms, delays, and complex non-linear processes described by differential equations. On the other hand, the AB and DE methods use a bottom-up perspective and deal with processes that are discrete and can handle middle to low levels of aggregation (i.e., low abstraction). The difference between these two methods resides in that the AB method deals with active and interactive evolving agents, whereas the DE method deals with passive and non-interactive agents. To the three specific modeling methods, one can add hybrid modeling tools that combine the best of each method [6,7].

As noted by Rahmandad and Sterman [8] regarding the AB and SD methods, one method is not better than the others; they are just different. Selecting the most appropriate method to model the dynamics of a given problem depends on "the purpose of the model and the level of aggregation appropriate for that purpose." In short, the selected level of aggregation must match the level of details in the available data sources and provide a balance between "simplicity and realistic depiction of the underlying mechanisms" expected to be at play in the problem of interest.

The purpose of this paper is to explore the applicability of the AB and SD methods to model a specific case study of managing water field services in a remote region (Afar regional state) of Ethiopia. The dynamic considered herein was adapted from that used in the Field Service AB model, which can be found in the Big Book of Simulation Modeling by Borshchev (pp. 142–194 [6]). The case study considered here involves multiple water borehole sites distributed over a given geographic area. The borehole units at each site serve the water needs of people in that area. The location of each site is defined by its latitude and longitude GIS coordinates. Each site involves a variety of equipment used for pumping (e.g., water pumps), storage (e.g., tanks), filtration (e.g., water filtration units), distribution (e.g., pipes, canals), and wastewater collection and treatment facilities, etc. When each borehole unit operates as planned, some water-related revenue stream is generated that benefits the population who depend on the borehole sites for livelihood and socio-economic development.

All borehole sites are managed by a single water utility that has adopted specific operation and maintenance rules and policies. A maintenance policy includes replacing site equipment after several cycles of maintenance. A policy is also in place to take care of any site equipment that needs to be repaired or replaced, assuming a specific rate of failure, which is expected to increase with equipment aging.

The water utility employs several motorized service crews (SC) that are initially all stationed at a single central location (service center) defined by its latitude and longitude GIS coordinates. When the utility receives a request of service (repair or replacement) from a borehole site, one of the service crews takes the request, drives to the site, and performs the necessary work of equipment repair, replacement, and/or maintenance. After completion of the work at each borehole site, the service crew takes another request from a request queue and drives to the next borehole site. If the queue is empty, it returns to the service center. All service crews are assumed to be able to communicate with each other and with a central dispatcher located at the service center. Furthermore, each service crew has fixed daily operating costs associated with its services. Each operation, such as repair, replacement, or maintenance, has an additional one-time charge.

In this paper, the AB and SD methods are used to model the dynamics outlined above. Both methods account, albeit in different ways, for the interaction of two groups of agents: the borehole sites and service crews. More specifically, the goal of this paper is to explore, for two different levels of abstraction, the capacity of each method to analyze how the management of the system of borehole sites and decision making over time are affected by (i) the location of the service center; (ii) the number of water utility service crews and their speed of transport, and (iii) the repair, replacement, and maintenance policy of the water utility. Two simulation modeling software tools are used in this paper: the agent-based modeling tool of AnyLogic (Personal Learning Edition, version 8.5.2) and the STELLA Architect system dynamics modeling tool by isee systems, Inc. (version 1.9.4.)

## 2. Agent-Based Simulation

A value proposition of agent-based modeling tools is to be able to capture the dynamic at play between multiple autonomous agents and heterogeneous groups of agents interacting in a system [9]. The agents have individual discrete behavior, and their interaction is defined by logic and a set of rules selected by the user. The interaction may change with time and location. Another advantage of the AB method is its ability to capture emergent phenomena at the global system level that may unfold through local agent interaction while considering their evolving behavior in a specific environment [6,10–13].

### 2.1. AB Model

The AB simulation modeling component of the AnyLogic software was used to model the dynamic between the two groups of agents considered in this case study, i.e., the borehole sites and the SC crews. The conceptual AB model used here is a modified version of the Field Service model proposed by Borshchev [6]. It has been modified to include a GIS map of a region of Ethiopia (Afar region) of interest, where 179 borehole sites are considered (Figure 2). The sites have specific GIS coordinates [14] and are spread over a 900 km long north–south stretch between Addis Ababa and Adigrat.

Table 1 lists the basic simulation parameters involved in the AB model. Note that other than the actual borehole site GIS coordinates used in this model, all other input parameters are, for demonstration purposes, virtually the same as those used in the Field Service model of Borshchev [6]. They do not reflect, however, the actual management of water field service in the region of Ethiopia under consideration. The AB model should be seen as a simplified version of the field reality with the sole purpose of being able to capture the dynamics at play between the borehole sites and the SC crews. It can also be understood as a template to be used in the management of water field services, or other services, used by communities in specific scales and contexts, once data are available.

The AB method can consider the various states that the two sets of agents may experience over a specific time, as well as possible state transitions. So-called "statecharts" are introduced to represent

these states, how agents enter the states and transition between states, and the decision process involved in these transitions.

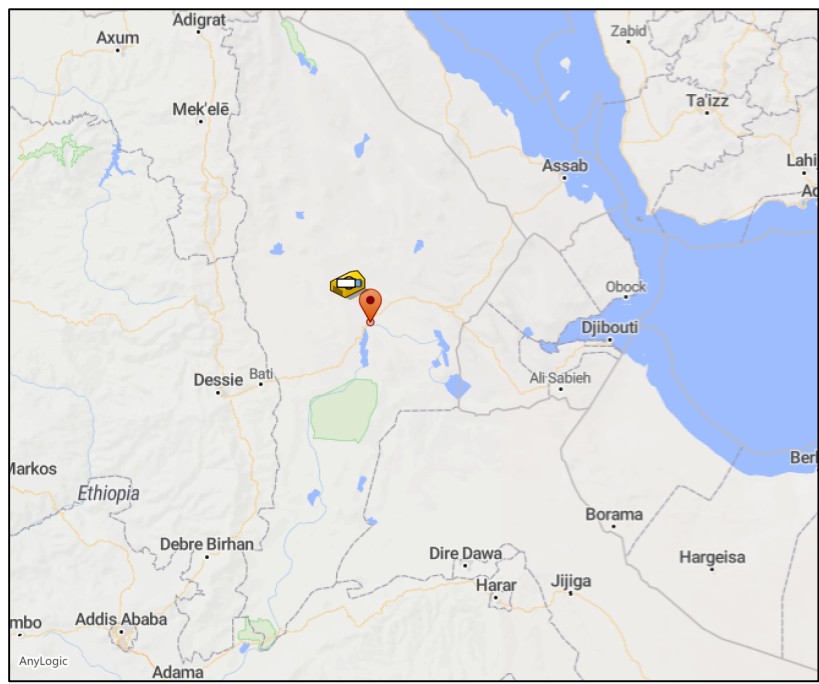

**Figure 2.** Map of the Afar region of Ethiopia considered in the agent-based (AB) model with the GIS location of the service center.

**Table 1.** Input parameters used in the agent-based model (adapted from Borshchev pp. 144, [6]).

| Parameters | Value |
| --- | --- |
| Service center location | GIS (latitude, longitude) |
| Number of borehole units | 179 |
| Borehole units location | GIS (latitude, longitude) |
| Number of motorized service crews (SCs) | 5 |
| Service capacity (number of SCs) | 5 |
| Equipment repair time (hours) | 5 |
| Equipment maintenance time (hours) | 3 |
| Equipment replacement time (hours) | 12 |
| Time between equipment maintenance (days) | 90 |
| Base equipment failure rate* (/day) | 0.03 |
| Probability of equipment replacement after failure | 0.1 |
| Number of maintenance periods before equipment is automatically replaced | 4 |
| Mobility of service crew (km/hour) | 10 (240 km/day) |
| Service crew daily operating costs ($/day) | 1500 |
| Average equipment repair one-time cost ($) | 1000 |
| Average equipment replacement one-time cost ($) | 10,000 |
| Average equipment maintenance one-time cost ($) | 600 |
| Daily revenue per working unit ($/day) | 400 |

* The actual failure rate increases as the equipment ages and has experienced several maintenance periods.

### 2.1.1. Equipment Statechart

Figure 3 shows the statechart used to map the lifecycle of each borehole site equipment which can go through five possible states over time: working (green), failed (red), replaced or repaired (yellow), or under maintenance (blue). The equipment at all borehole sites is assumed to be initially working (green). The transition from one state to the next can be triggered by (i) a state change rate

(e.g., failure rate); (ii) a timeout (e.g., time to finish maintenance, replacement, or repair) or delay; or (iii) a message between agents (e.g., SC arrived for repair, SC arrived for maintenance). In Figure 3, places of decision (equivalent to "if" statements) are represented by three lozenge-shaped symbols. One choice (probabilistic) may be to repair or replace the equipment when an SC arrives at a site where the equipment failed; repair is the default. The equipment is automatically replaced when its age is such that it has experienced at least a certain number (4 in the case of Table 1) of consecutive maintenance periods (90 days). After the repair is done, a decision (deterministic) needs to be made whether the equipment is already due for regular maintenance (every 90 days); no maintenance is the default. Finally, another deterministic decision (planned replacement) when an SC arrives at a site to carry out regular maintenance (every 90 days) is whether to proceed or replace the equipment; maintenance only is the default. As in the case of failed equipment, replacement automatically takes place if its age is such that it has experienced at least a certain number (4 in the case of Table 1) of consecutive maintenance periods of 90 days.

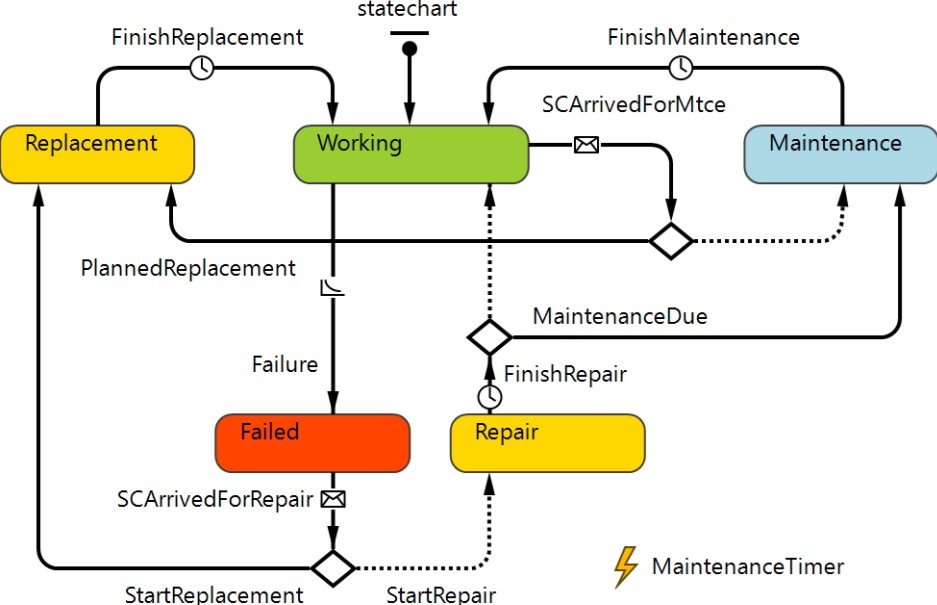

**Figure 3.** Statechart showing the different possible states of the borehole units, the transitions between states, and the places of decision (lozenge-shaped symbols) (from Borshchev pp. 180, [6]).

### 2.1.2. Service Crew Statechart

Figure 4 shows the statechart for the motorized service crews (SC), which can be in four different states: idle at the service center (red), working (yellow), or on the move either to the service center (waiting for a message from a borehole site), or to a borehole site for servicing (orange). The service crews transition from one state to another after exchanging different types of messages: (i) from each borehole site (person or signal) to the service crews (e.g., check request messages) via the service center (e.g., request for service message); (ii) from the service crews to the service center (e.g., accept message to proceed or re-check for a message and take it); (iii) from the service crews to the borehole sites (e.g., arrived message); and (iv) from the borehole sites to the service crews (e.g., finished message that the service is completed).

In Figure 4, places of decision (equivalent to "if" statements) are represented by two lozenge-shaped symbols. One decision after checking the request queue sent from the service center is for each SC whether to drive to the worksite or return to the service center (driving home). Another decision is to check whether each service crew is still needed for service (e.g., still employed or laid off). This decision is controlled by the modeler using a service capacity parameter, which describes the number of SC units deemed necessary for service. If, at any time, the service capacity is larger than the current

number of SC units, new SC units are added to close the gap. If the service capacity is smaller than the current number of SC units, some are laid off, as indicated in Figure 4. In other words, the service capacity always represents the default value for the number of service crews.

Furthermore, all service crews are assumed to follow specific operative and mobility rules (p. 188, [6]). The service is provided 24 h a day, and all service crews "always have all the necessary parts and tools on board. They never have to drive to the base location [service center] to pick up missing stuff." Since the present model uses a GIS map, all service crews follow roads and trails on the map. Finally, the service crews do not seek the closest borehole sites once they receive a request from the service center. They "just take the next request from the queue and drive there." These operative and mobility rules could be overcome, in part, by combining the agent-based method with the discrete-event method discussed in Section 1.

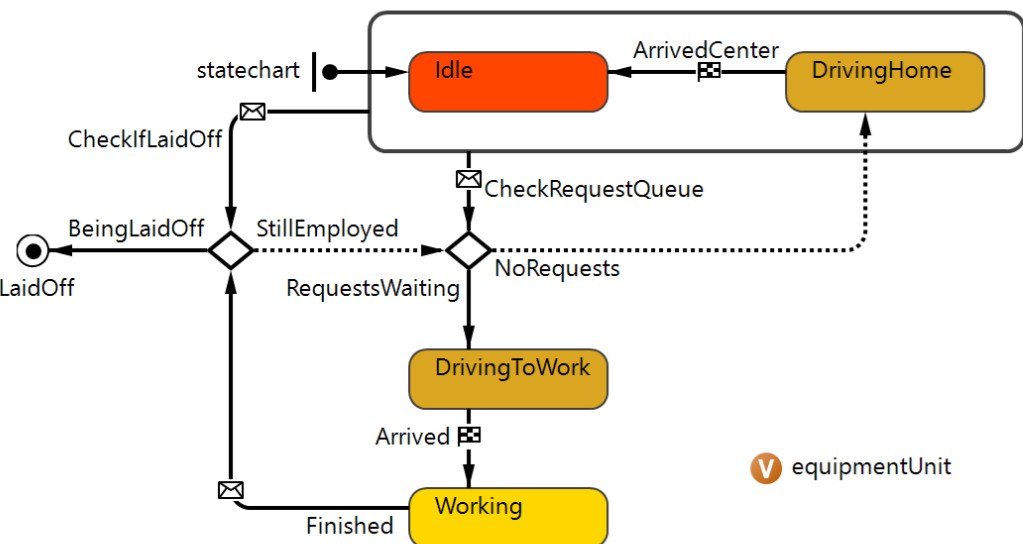

**Figure 4.** Stage chart showing the different possible states of the service crews, transitions between states, and decision places (lozenge-shaped symbols) (from Borshchev [6], p. 179).

*2.2. Flight Simulator*

2.2.1. Example

The agent-based Field Service model from the Big Book of Simulation Modeling by Borshchev [6] comes with a flight simulator consisting of two parts: (i) a two-dimensional (2D) model (Figure 5) showing the agent dynamics (movement of the service crews and states of each borehole site) and (ii) a metrics and statistics window showing time-depend graphs (Figure 6). The model FieldService2DEthiopia used here is available from the author upon request.

Figure 5 is a snapshot of the 2D model showing the status of the equipment for the 179 borehole sites and the movement of the service crews at a specific model simulation time. The trucks representing the service crews are assumed to follow the roads and trails on the GIS map of Figure 2.

Figure 6 shows the daily variation of the state of the equipment and the change over five years (1825 days) of the annual averages of (i) equipment availability, (ii) service crew utilization, and (iii) cost, revenue, and profit involved in the operation and maintenance of the equipment at the borehole sites. After about two years (730 days), the fractions of working equipment and working and driving service crews increase slightly with time. This trend also translates into a slightly increasing annual cost and slightly decreasing yearly revenue and profit. Overall, Figure 6 shows that a service capacity consisting of five service crews is not enough to keep up with the failed equipment; hence, the service capacity must increase.

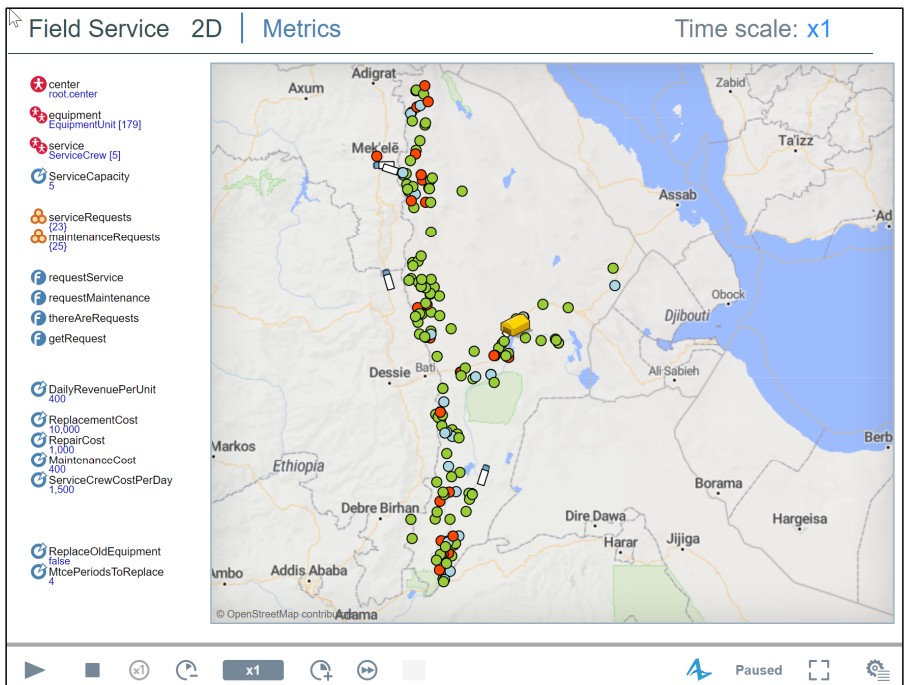

**Figure 5.** 2D model part of the AB flight simulator in AnyLogic showing the fixed service center and the status of the equipment at the 179 borehole sites using the color code of Figure 3.

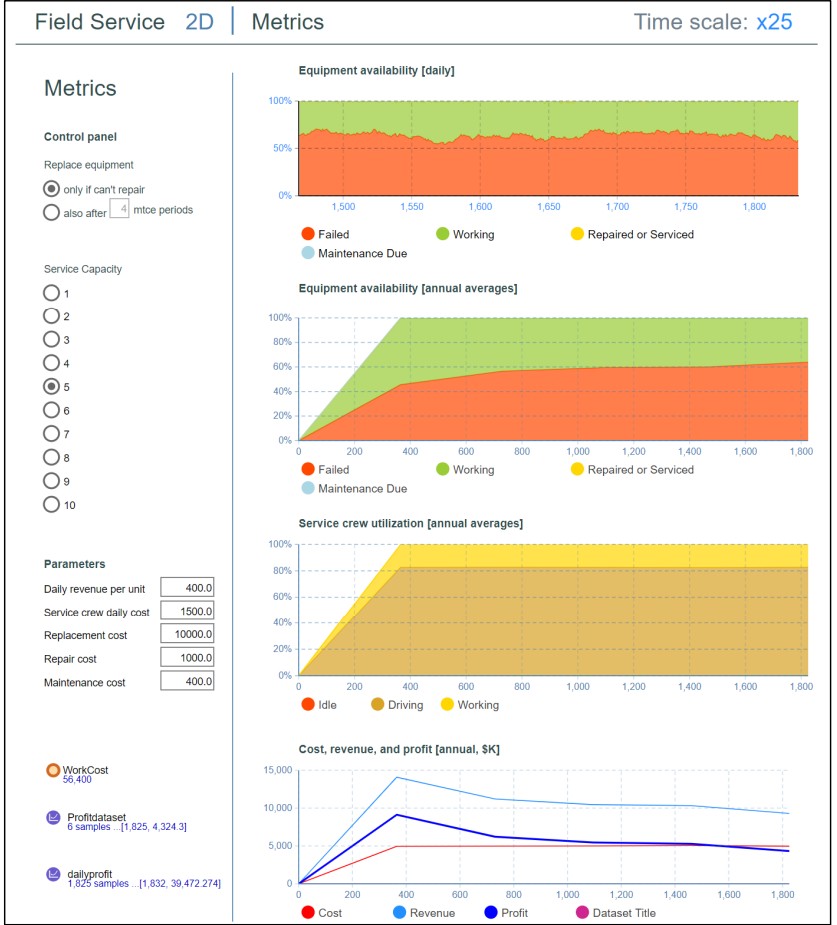

**Figure 6.** The second part of the AB flight simulator in AnyLogic showing (i) daily and annual average equipment availability, and (ii) annual average service crew utilization, cost, revenue, and profit.

2.2.2. Effect of Service Capacity

A unique feature of the flight simulator shown in Figure 6 is the possibility of changing at any time: (i) the number of service crews using the service capacity parameter (from 1 to 10 in the present example), (ii) the daily revenue per unit, and (iii) the service crew, replacement, repair, and maintenance costs. Finally, the user can decide whether the borehole equipment is automatically replaced if it cannot be repaired, or if it has also experienced an arbitrary number of consecutive maintenance cycles of fixed duration (four cycles of 90 days each in the present example). Both decisions were discussed in Section 2.1.1.

Figure 7 shows the variation over five years of the annual average cost, revenue, and profit involved in the operation and maintenance of the equipment at the borehole sites when the service center capacity increases by one service crew at the end of each year from five initially to ten at the end of year 5 (1825 days). Table 2 compares the annual average profit assuming a constant number of five service crews with that assuming an increase of one service crew per year. The results clearly show the sensitivity of the annual profit on the number of service crews. A yearly increase in service crews, from five in year one to ten in year five, results in increasing the cumulative five-year profit by 53%.

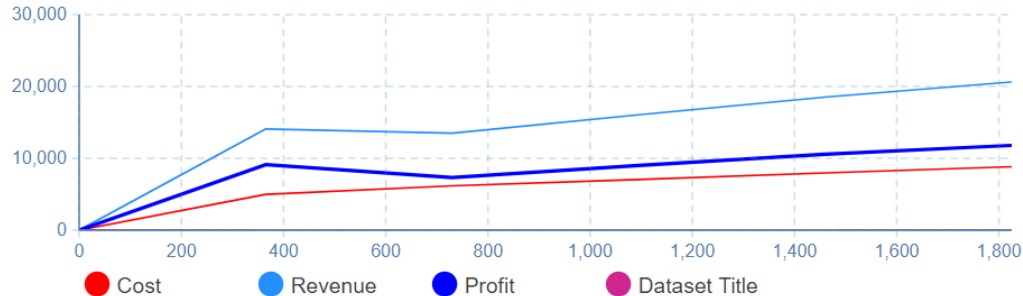

**Figure 7.** Variation of annual cost, revenue, and profit over five years, assuming an increase of one service crew (SC) at the end of each year.

**Table 2.** Comparison of annual average profit (k$) and five-year cumulative profit assuming a constant number of five service crews (SC) with that assuming an increase of one service crew per year starting with five of them.

| Years | Annual Average Profit (k$) (Constant SC) | Annual Average Profit (k$) (Increasing SC) |
|---|---|---|
| 1 | 9123 | 9123 |
| 2 | 6227 | 6890 |
| 3 | 5465 | 8725 |
| 4 | 5283 | 10,612 |
| 5 | 4324 | 11,169 |
| Cumulative | 30,422 | 46,519 |

2.2.3. Optimization

The Field Service Model of Borshchev (p. 182, [6]) also contains a built-in optimizer (OptQuest^{TM} optimizer), which can be useful in exploring how some of the parameters listed in Table 1 have more impact than others on the lifecycle of the equipment at the 179 borehole sites. We might be interested in determining, for instance, what combination of parameters leads to a maximum profit over a certain period. For the present example, the objective function Profit (t) is defined as follows:

$$Profit\ (t) = (Mean\ Daily\ Revenue - Daily\ Service\ Crew\ Cost) * t - Total\ Work\ Cost \qquad (1)$$

where $t$ is the time expressed in days. The mean daily revenue is calculated as the product between the mean number of borehole units working daily and the daily revenue per working unit ($400 in Table 1). The daily service crew cost is the product of the number of daily working service units and the service crew daily operating costs ($1500 in Table 1). The total work cost is the accumulated cost associated with the incremental operation (repair, replacement) and maintenance costs of the borehole units over time.

As a numerical optimization example, let's consider the case where the number of maintenance periods before the equipment is replaced varies from 1–5, and the service capacity varies from 3–10 service crews. For this example and at the end of five years (1825 days), the optimization algorithm gives an optimal annual profit of $41,930 (k$) for a one-time maintenance cost of $400; four maintenance periods of 90 days; and service capacity of six service crews.

### 2.2.4. Influence of Service Center Location

A parameter likely to influence cost and profit is the location of the service center relative to the fixed borehole sites. Four possible locations of the service center were considered on the map of Figure 2 to analyze that effect: the position shown in Figure 2, Mek'ele Dessie, and Addis Ababa. Table 3 summarizes the impact of the service center's location on the annual average profit and the cumulative profit over five years. The location of the service center does not seem to have much of an influence on the cumulative profit at the end of five years.

**Table 3.** Comparison of annual average profit (k$) and five-year cumulative profit for four different locations of the service center, all other input parameters being the same.

| Year | Figure 2 11.74506, 41.00237 | Mek'ele 13.50247 39.47589 | Dessie 11.12075, 39.61866 | Addis Ababa 9.01238, 38.76209 |
|------|------|------|------|------|
| 1 | 9123 | 8857 | 8910 | 9459 |
| 2 | 6227 | 5904 | 6323 | 5834 |
| 3 | 5465 | 5782 | 5529 | 5510 |
| 4 | 5283 | 5321 | 5653 | 4975 |
| 5 | 4324 | 4809 | 4989 | 4918 |
| Cumulative | 30,422 | 30,673 | 31,404 | 30,696 |

An optimization analysis, like the one mentioned in Section 2.2.3, was carried out for the four locations of the service center. The results listed in Table 4 show that, if optimizing the cumulative profit at the end of five years (1825 days) is the goal, the different service locations yield similar values of the profit but require slightly different operation and maintenance policies in terms of the number of 90 day maintenance periods before the equipment is replaced and the number of service crews.

**Table 4.** Optimization scenarios for four different locations of the service center, all other input parameters being the same. Time window of five years.

| Service Center Location | Five-Year Profit (k$) | Maintenance One-Time Cost ($) | Number of Maintenance Periods before Equipment is Replaced | Capacity Service. Number of Service Crews |
|------|------|------|------|------|
| Figure 2 | 41,930 | 400 | 4 periods of 90 days | 6 |
| Mek'ele | 40,910 | 400 | 5 periods of 90 days | 10 |
| Dessie | 39,998 | 400 | 4 periods of 90 days | 6 |
| Addis | 41,703 | 400 | 4 periods of 90 days | 8 |

### 3. System Dynamics Simulation

The system dynamics method is a relatively recent branch of systems science that originated with the work of Dr. Jay Forrester at the Massachusetts Institute of Technology in the 1950s and 1960s [15–17]. A review of the method and its multiple applications can be found in some landmark books by Richmond [18], Sterman [19], and Ford [20], among many others. The unique characteristics of the SD method that warrant its use in modeling the dynamics of complex systems include being able to (i) study how systems continuously change over time due to possible changes in and relationships among components and changes in the overall direction of systems allowing for both qualitative and quantitative modeling; (ii) account for systems non-linearities, feedback mechanisms, and delays; (iii) illustrate that as the structure of a system changes, so does its behavior and vice-versa. A limiting aspect of SD is that it cannot capture the details of the individual components that form the system.

*3.1. SD Model*

Compared to the AB method, system dynamics (SD) considers all borehole equipment sites and service crews as two aggregated and indistinguishable homogeneous sets of agents that are evolving continuously. As a result, this abstraction level makes the modeling of how the agents in Figures 3 and 4 interact and change more challenging. For instance, it is not possible to capture the individual mobility of the service crews directly as the borehole equipment sites transition from one state to the other. Only the average behavior of the two sets of agents can be estimated. One way to overcome some of these limitations is to carry out sensitivity and stochastic analyses on key parameters in the SD models, as shown below.

Having this in mind, Figure 8 shows a possible stock-and-flow system dynamics simulation model that captures as best as possible the dynamic between the different states in the equipment statechart of Figure 2. The model was developed by the author using the STELLA Architect software (version 1.9.4). A user interface of the system dynamics model is available on the web (https://exchange.iseesystems.com/public/bernardamadei/bhserviceethiopia) and can be used to explore different scenarios. A special effort was made to integrate the same data (Table 1) used in the AB model in the SD simulation.

In the SD model of Figure 8, several stocks are used to represent the state of the equipment at the borehole sites. Different states (e.g., working, repaired, failed, replaced, and maintained) and transitional states (e.g., failed, needing replacement, and needing maintenance) are considered. Three decision time converters, expressed in hours, are used to capture the time necessary for the borehole units to return from a repair, maintenance, or replacement state to a working state. The three decision times are selected at random, assuming a triangular distribution ranging between 12 and 48 h with a mode of 24 h.

The equipment failure rate is represented by a time function specified by the user. Since it was not possible to reproduce with SD the failure rate algorithm used in the AB model, an S-shaped function was selected to match as best as possible the variation of the failure rate considered in that model. The failure rate varies between a minimum value (0.03 in this example) and a maximum value specified by the user (0.11 in this example). The slope of the S-shaped function (0.01 in this example) dictates how quickly the failure rate increases with time. If necessary, the maximum value and slope can themselves be made dependent on specific maintenance rules and policy selected by the utility company.

After failure, some borehole sites have their equipment repaired, while others require equipment replacement. The basic rate of equipment replacement after failure is equal to 10% per day (see Table 1). Once the equipment is replaced after a specific time, the borehole units are returned to a working state. If not replaced, the equipment is repaired. It may happen that during repair, the equipment is due for maintenance. If not, the borehole units are returned to a working state. If yes, maintenance of the equipment is performed, and the borehole units are returned to the working state. Finally, borehole units that follow a regular maintenance policy are checked whether they have exceeded a specified

number (four) of cycles of maintenance (90 days each). If yes, their equipment is automatically replaced. If not, the borehole units are returned to a working condition.

The daily number of service crews necessary to provide the replacement, repair, and maintenance services is calculated as the ratio between the daily number of boreholes repaired, replaced, or maintained calculated by the model and the number of borehole units assumed to be serviced by each service crew. The latter is selected at random, assuming a triangular distribution ranging between three and seven with a mode/mean value of five borehole units served by each service crew.

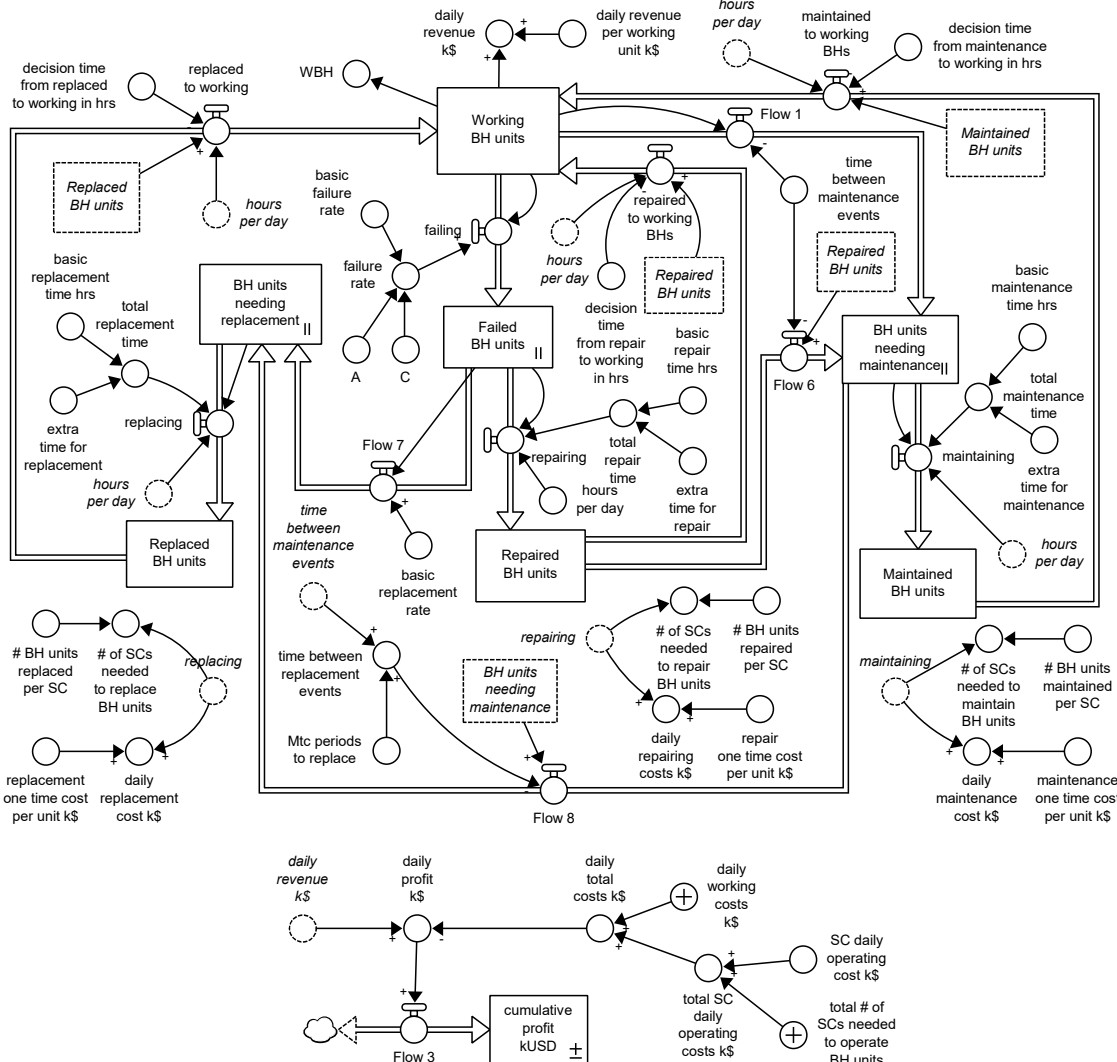

**Figure 8.** Stock-and-flow system dynamics (SD) model showing the dynamic between the different borehole equipment states.

It should be noted that the travel time of the service crews to each borehole site is not accounted for explicitly in the SD model, like in the AB model. As an alternative, the travel time is assumed to be random, assuming a triangular distribution ranging between 15 and 45 h with a mode/mean value of 30 h. This extra time is added to the basic equipment repair time (5 h), maintenance time (3 h), and replacement time (12 h) of Table 1. The selected range of travel time translates into a travel distance ranging from 150 to 450 km at a speed of 10 km/hour (see Table 1), which corresponds somewhat to the range of travel distances expected from the service center to the northern limit (Adigrat) and southern limit (Addis Ababa) of Figure 2. Recall that the distance between these two limits is about 900 km and that the service center is located roughly halfway (about 450 km) in between.

As an example, Figure 9 shows the variation of the cumulative revenue, cost, and profit over five years for the input data mentioned above. The cumulative profit predicted by the AB method (listed in Table 2) is also shown for comparison.

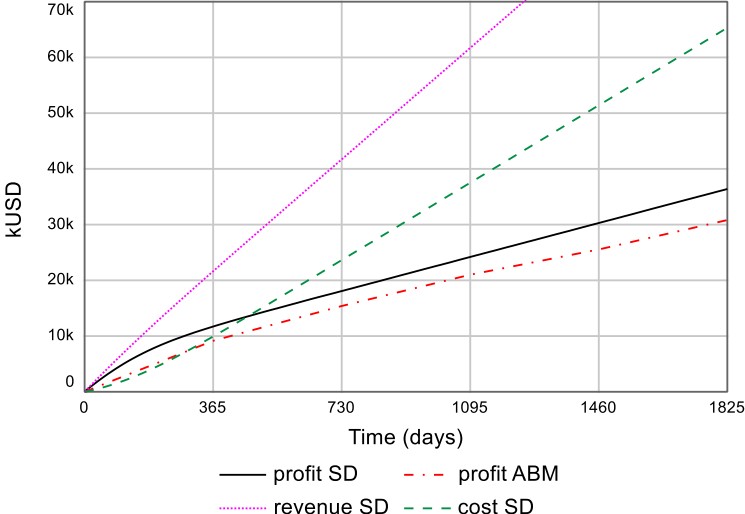

**Figure 9.** Cumulative revenue, cost, and profit over five years (1825 days). The cumulative profit calculated using the AB method is shown for comparison.

### 3.2. Sensitivity Analysis

The SD model presented above can be used to carry out various sensitivity analyses. The baseline values of the parameters involved in the analyses presented below were selected as follows: (i) time between maintenance events = 90 h; (ii) extra times for service (repair, replacement, maintenance) = 30 h; (iii) decision times from service (repair, replacement, maintenance) to working = 48 h; and (iv) number of borehole units serviced (repair, replacement, maintenance) per service crew = 5. The sensitivity analyses consisted of varying each one of these parameters incrementally over specific ranges of values.

### 3.2.1. Influence of Time between Maintenance Events

As an example, Figure 10 shows the influence of the time between two consecutive maintenance events on the performance of the system. That time is assumed to vary between 20 and 200 days, all other parameters being the same. Figure 10 shows a significant influence after 365 days for a 100% confidence interval. For smaller levels of confidence, the impact is limited.

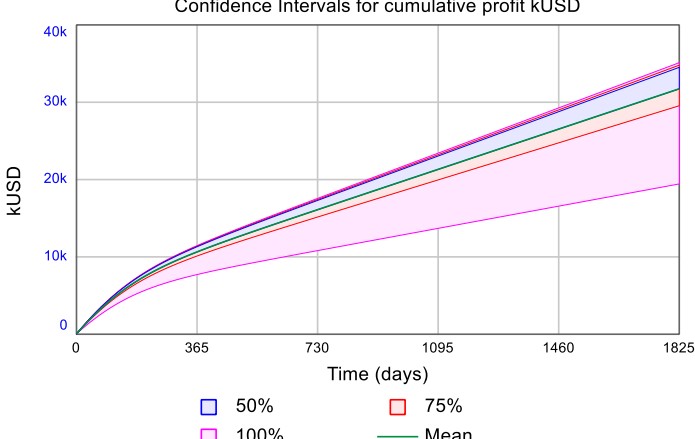

**Figure 10.** Cumulative profit over five years (1825 days) for different values of the time between two consecutive maintenance events. The mean value and confidence intervals are shown.

### 3.2.2. Influence of Extra Service Time

A second sensitivity analysis was carried out to explore the effect of the extra times for repair, replacement, and maintenance on the cumulative profit. All three times were assumed to vary between 15 and 45 h, all other parameters being the same. The sensitivity of the profit to the extra times after 365 days is clearly emphasized in Figure 11.

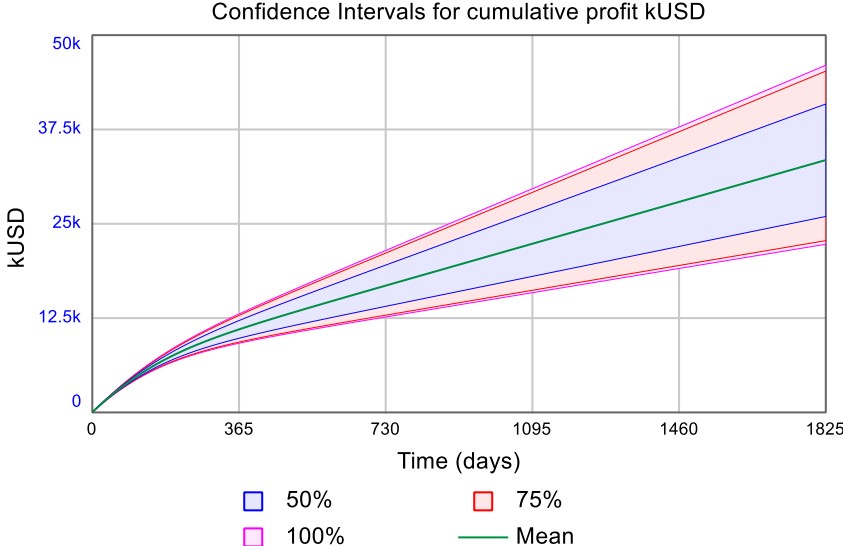

**Figure 11.** Cumulative profit over five years (1825 days) for different values of the extra times for repair, replacement, and maintenance. The mean value and confidence intervals are shown.

### 3.2.3. Influence of Decision Time between Service and Working

A third sensitivity analysis was carried out to explore the effect of the three decision times in Figure 8 (repaired to working, replaced to working, and maintained to working) on the profit. All three decision times were assumed to vary incrementally between 24 and 96 h each, all other parameters being the same. As shown in Figure 12, the decision time has a limited influence on the cumulative profit.

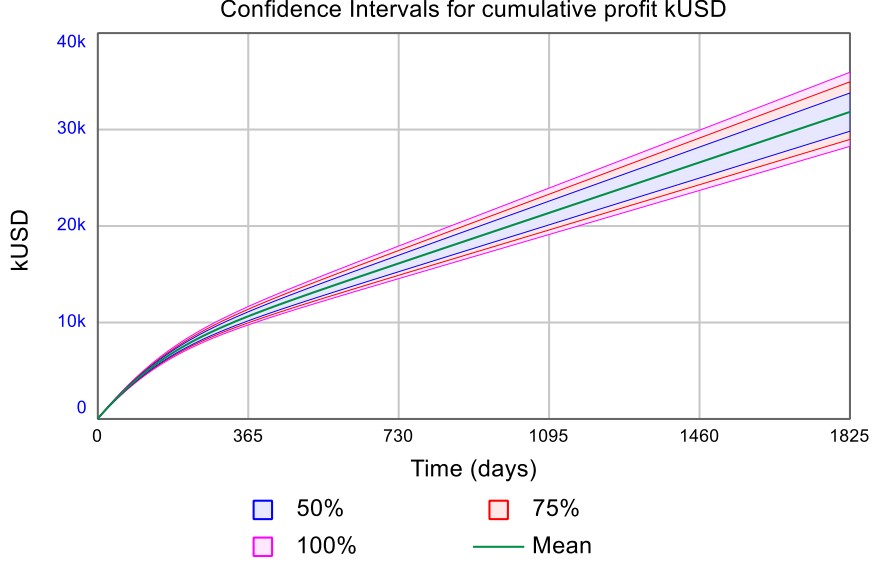

**Figure 12.** Cumulative profit over five years (1825 days) for different values of the decision times (repair to working, replaced to working, and maintained to working). The mean value and confidence intervals are shown.

### 3.2.4. Influence of Number of Borehole Units per Service Crew

The last sensitivity analysis explores the influence that the number of boreholes serviced per crew has on the cumulative profit. That number is assumed to vary incrementally between three and seven for repair, replacement, and maintenance; all other parameters being the same. The results of the analysis shown in Figure 13 show that the number of boreholes serviced per crew has a limited effect on the profit for the range considered herein.

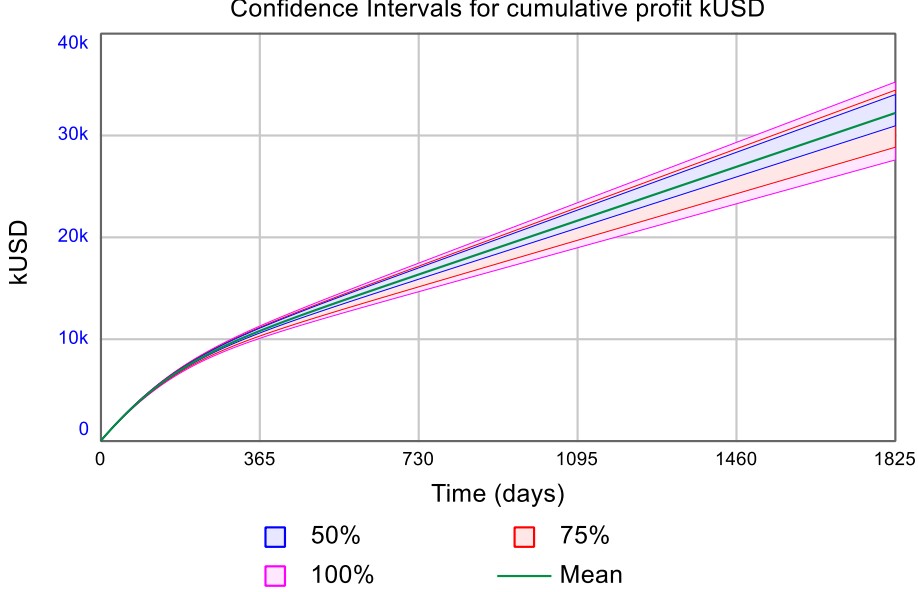

**Figure 13.** Cumulative profit over five years (1825 days) for different values of the number of borehole units per repair, replacement, and maintenance crew. The mean value and confidence intervals are shown.

### 3.3. Optimization Analysis

The STELLA Architect software can also be used to carry out optimization work using different algorithms such as the Powell, grid, and differential evolution methods [21]. As an illustrative example, the differential evolution algorithm was used to determine the values of decision times and extra times that lead to the maximum cumulative profit at the end of five years (1825 days). All other model parameters were assumed to be the same. An optimal profit of $50,408 (k$) was found for the following input values:

- Decision times between repair, maintenance, or replacement and working: 24 h
- Extra times for repair, replacement, and maintenance: 15 h

For these conditions, the management of the borehole sites can be accomplished by changing the number of service crews providing the services of repair, replacement, and maintenance. In this example, each service crew is assumed to serve five borehole units per day. Figure 14 shows the variation of the total number of service crews with time.

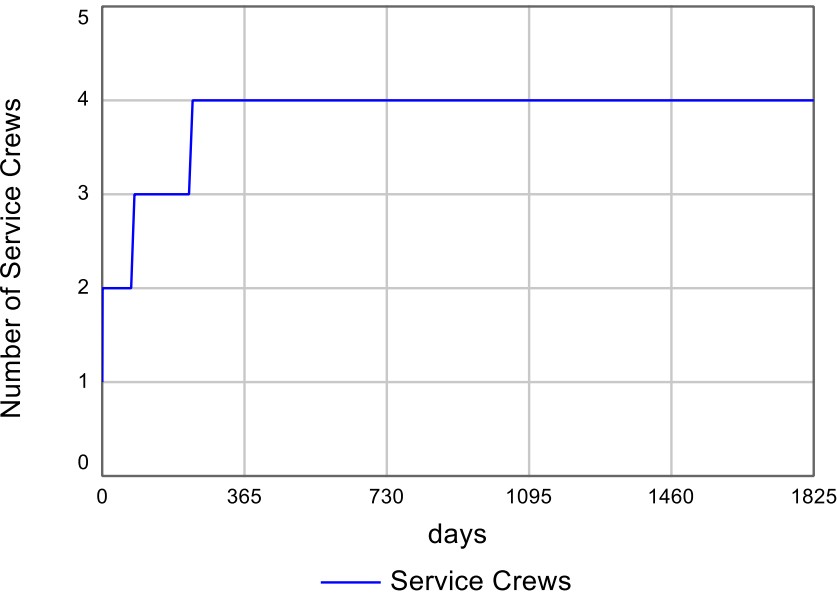

**Figure 14.** Variation of the number of service crews with time.

## 4. Conclusions

This paper looked at the application of two numerical methods to model the dynamic between two groups of interacting agents involved in the management of water field services in a remote region of Ethiopia. Both methods use different formulations and levels of abstraction to model that interaction.

The disaggregation inherent to the AB method allows for a detailed and discrete analysis of the impact individual agents and groups of agents have on the performance of complex systems. This unique characteristic of the AB method allows for capturing emergence, a phenomenon that is unique to complex systems and which cannot be obtained using linear equations, or the SD and DE methods. In the case study considered herein, the interaction of the borehole site and service crew agents creates an emerging behavior that cannot be inferred from the sum of individual interactions alone.

The SD method assumes a high level of abstraction and homogeneity when considering the interaction between the two groups of agents. The method requires fewer details than the AB method and uses an averaging process to predict the continuous behavior of the two groups of agents. Various parametric studies can still be carried out to analyze, for instance, the impact of multiple factors on the cumulative profit. However, the influence of the location of the service center on the profit cannot be analyzed.

The two AB and SD modeling software tools used in this paper are powerful enough to handle multiple complex processes and interactions. Despite their differences and for the case study considered in this paper, both AB and SD modeling tools give results that are realistic, good enough, and consistent in the ballpark with each other. Of course, many assumptions would have to be made to match the predictions of both methods even further.

As emphasized in the literature comparing AB and SD methods, one approach is not better than the other [11,22]. They are different in the way they interpret reality and can be used for various aspects of decision making in addition to matching the level of disaggregation in the situation of interest. A metaphor used by Schieritz and Milling [7] to describe the main difference between the two methods is that the SD method models the trees, and the AB method models the forest. The decision to use one method rather than the other should be based on whether decision-makers interested in exploratory policies are involved at the strategic level, where SD is more appropriate, or the operational level where AB is more relevant. In some situations, the value proposition of both approaches (i.e., modeling the forest and the trees) should be considered when considering how strategy translates into operational work.

Despite being arbitrary selected, the numerical example used in this paper shows that the different complex steps involved in the management of water field services can be captured explicitly using AB and SD models. As emphasized in Section 1, in their interpretation of reality, the models can be useful to "reason, explain, design, communicate, act, predict, and explore" [2]. It must be kept in mind that models are as good as the data available from field studies.

Finally, it should be noted that the AB and SD models presented above do not specify the sequences of operations and processes taking place at each borehole site from the arrival of a service crew to its departure once repair, replacement, and maintenance are completed. The operations may include, for instance, crew assessment, sequential repairing, and replacement of some of the borehole equipment, finding missing parts, etc. This additional complexity would require another level of disaggregation, which, if needed in operation and maintenance, could be accounted for by combining either method with the discrete-event (DE) method mentioned earlier in this paper.

**Author Contributions:** Author is the sole contributor of this paper.

**Funding:** This research received no external funding.

**Conflicts of Interest:** The author declares no conflict of interest.

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
