# Peer review of "Agent-Based and System Dynamics Modeling of Water Field Services"

_challenges, doi:10.3390/challe11020013_

Round 1

Reviewer 1 Report

Figure 2 Author should add legend

Figure 5 Author should add legend

In general the analysis use ideal world and therefor results will be far away from reality. Use of Etiopia example didn't provide any realistic ratings of proposed methods.

Author Response

Figure 2. Legend now reads as follows: " GIS map of the region of Ethiopia considered in the ABM model with the location of the service center."

Figure 5. Legend is already there. Change "Figure 2" to "Figure 3" in the legend. 

Yes, this is an ideal world to show how two different methods can be used to model it. As mentioned in lines 36-42 the model is an interpretation of reality and is not reality itself. Also, please read the disclaimer from lines 118-122. 

Reviewer 2 Report

Starting with the observation mentioned at lines 39-40 will be really useful to comment a little bit the real situation described at point 2.1. Could be in conjunction with some additional observations related to Field Services model of Borshchev (eg - lines 121-122 to be detailed).

Titles 2 and 3 should be reformulated, more specific to the task, in the framework of own methods (AB and SD) possibilities to model the given situation.

At the conclusion section are necessary to be inserted some explanations related to interoperability of AB and SD models, highlighting reasons for selected them and not others.

Finally, would be useful for the soundness to provide some lines related to possibilities to use the research and discussions for others similar cases in water field services.

Author Response

1. Statement added

2. I reformulated the titles 2 and 3 as "Agent-based formulation" and "System Dynamics formulation."

3. I think I addressed that remark (lines 421-430)

4. The recommended statement was included after line 122 and as a sentence in the conclusion

Reviewer 3 Report

I want to thank the author to had the opportunity to read this work.

The article is very interesting, well stractured and the English is very good.

I not have major comments, but only minor suggestions:

  1. Figure 2. It is possible to add a legend?
  2. line 175. "They work 24 hours a day". Is this real? or it is an assumption of the model?
  3. line 175-176. I think that this can be a strong assumption in the model. Maybe can be better investigated considering an higher time for the repair, that must be statistical based on real data. In this way should be possible to consider also the time needed to "search" the missing parts.

Author Response

  1. Legend was inserted. 
  2. I meant "service is assumed to be provided 24 hours a day." 
  3. This additional level of complexity could not be handled by the ABM and SD methods. As mentioned in the last paragraph of the conclusion, the discrete event method would have to be used. I have added a sentence at the end of line 179. "All these operative and mobility rules can be overcome, in part, by combining the AB method with the discrete-event (DE) method discussed in the introduction."